# Breeding Behavior, Nestling Growth, and Begging Behavior in the Plain Laughingthrush (*Garrulax davidi*): Implications for Parent–Offspring Conflict

**DOI:** 10.3390/ani13223522

**Published:** 2023-11-15

**Authors:** Jinyuan Zeng, Yueqi Li, Long Zhao, Yurou Shi, Saba Gul, Hongquan Shi, Sen Song

**Affiliations:** 1School of Life Sciences, Lanzhou University, Lanzhou 730000, China; zengjy20@lzu.edu.cn (J.Z.); liyueqi3068@163.com (Y.L.); shiyr21@lzu.edu.cn (Y.S.); saba.swabi@yahoo.com (S.G.); 2Gansu Gahaizecha National Nature Reserve, Gannan Tibetan Autonomous Prefecture 747200, China; zhaolong868@126.com; 3Gansu Key Laboratory of Protection and Utilization for Biological Resources and Ecological Restoration, Qingyang 745000, China

**Keywords:** *Garrulax davidi*, parental feeding strategy, sibling rivalry, nestling begging, re-feeding tactic

## Abstract

**Simple Summary:**

Food allocation among nestlings of altricial birds is crucial for understanding the evolution of parent–offspring conflicts within avian families. Empirical studies have yet to reach a consensus on whether parents or offspring determine the food distribution within the brood. In the case of the Plain Laughingthrush (*Garrulax davidi*), we explore the relationship between parental feeding strategies and nestling begging behaviors. Due to hatching asynchrony, larger nestlings often outcompete their smaller siblings for food, although they do not consistently exhibit higher begging intensity. Generally, nestlings with the highest begging intensity are more likely to be fed first, underscoring the importance of nestling begging in parental food allocation. However, if the initial food recipients are already satiated and do not immediately consume the food, parents reallocate it to other nestlings. This re-feeding tactic reduces the chance of early-hatched nestlings monopolizing food due to their larger size. Our research demonstrates that, while parental food allocation primarily depends on nestling begging intensity, the decision to re-feed hinges on whether the initial recipients promptly ingest the food.

**Abstract:**

Investigation on food allocation among nestlings of altricial birds is crucial in understanding parent–offspring conflicts within avian families. However, there is no consensus in empirical studies regarding whether parents or offspring determine the food allocation pattern within a brood. In the Plain Laughingthrush (*Garrulax davidi*), we examine the relationship between parental feeding strategies and nestling begging behaviors. Due to hatching asynchrony, larger nestlings have a competitive advantage in food acquisition over their smaller brood-mates; nevertheless, if the initial food-receivers were already satiated and did not immediately consume the food, parents would retrieve the food and re-allocate it to another nestling. This re-feeding tactic employed by parents reduced the likelihood of early-hatched nestlings monopolizing the food solely due to their larger body size. Our findings indicate that parents primarily allocated food based on nestling begging intensity, while their re-feeding tactic is determined by whether the first food-receivers have consumed the food. To date, our research demonstrates that while parental food allocation primarily hinges on the begging intensity of the nestlings, the decision to re-feed is contingent upon whether the initial recipients of the food ingest it immediately.

## 1. Introduction

In 1974, Robert Trivers proposed the concept of parent–offspring conflict (POC) to describe the interplay between parents and offspring in the allocation of parental investment [1]. Since parents are equally related to each of their offspring, they are expected to balance the investment among all their offspring. However, an individual offspring is fully related to itself but only half related to its siblings. As a result, offspring have a tendency to monopolize parental investment, regardless of the costs imposed on their siblings [1,2,3]. The concept of POC has revolutionized studies of animal behaviors [4,5,6,7], as well as quantitative genetics [8,9,10]. The question of who determines the food allocation pattern within the brood has long been addressed for altricial birds with multiple young in one breeding attempt [1,5,11]. Given the dominance of adults to nestlings, it is intuitive that parents control the food allocation among nestlings. For example, parents may determine which nestling receives more food than others [12,13,14,15,16,17,18], determine the size and amount of food items delivered to the nest [19,20], or dictate the feeding position when arriving at the nest edge [21]. However, empirical studies have revealed that dependent young can also influence the food allocation pattern, mainly through sibling rivalry involving behaviors such as mutually pecking, jostling, or exaggerated begging [15,22,23,24]. Especially, the large-sized offspring that hatched early may have an advantage in this rivalry. To date, elucidating the relative contributions of parents and offspring in determining food distribution remains a crucial endeavor for understanding the evolution of parent–offspring conflict (POC) in animal families [25,26,27,28].

When birds produce large clutches, they often begin incubation before the entire clutch is laid, resulting in asynchronous hatching. This leads to the emergence of early-hatched nestlings that are larger and more competitive than their younger siblings [15,29,30,31]. Once a size hierarchy is established among the nestlings, the feeding strategy of the parents will determine the fate of the different-sized nestlings. If parents provide more food to larger nestlings, these favored nestlings may outcompete the smaller ones, potentially leading to starvation among the latter [32,33,34]. In the other hand, if parents provide more food to the smaller nestlings, it may partially compensate for the disadvantage of their smaller size, allowing them to catch up with the growth of the larger nestlings [18,35]. Nestlings often exaggerate their begging behaviors to attract parent attention, rather than solely reflecting their actual need for food [36]. In this context, parents need to discriminate between cues of genuine hunger and signals driven by sibling rivalry. This ability allows parents to enhance their reproductive success by allocating resources appropriately in the current environmental conditions [37,38]. Failing to discriminate between genuine need and dishonest begging signals can be costly for parents, as intense sibling competition may lead to brood reduction [39]. Currently, the mechanisms by which parents are able to differentiate between genuine and dishonest signals of need in nestling begging behaviors has not been fully resolved. To demonstrate the evolution of POCs in birds, three important questions need to be addressed: (i) Are there any differences in growth rate among chicks in nests with asynchronous hatching? (ii) How do parents distinguish between genuine and dishonest begging signals from different-sized chicks? (iii) How do parents respond to different begging behaviors exhibited by the chicks when provisioning the brood?

We sought these questions in the Plain Laughingthrush *Garrulax davidi*, a middle-sized thrush that is exclusively endemic to China but distributes widely in the north of China. These birds are known to inhabit the northward region up to the Loess Plateau and the westward region up to the Tibetan Plateau, which is characterized by an altitude of over 3000 m. This bird species is further classified into three subspecies: *G. davidi concolor*, *G. davidi davidi*, and *G. davidi experrectus*. Unfortunately, there have been only two reported studies on their breeding biology [40]. Among these studies, one specifically focused on the subspecies *G. davidi experrectus*, which breeds on the Tibetan Plateau. The findings of this study revealed that breeders of the subspecies produce significantly larger clutch sizes compared to those breeding in low-altitude regions. In the breeding behavior of the Plain Laughingthrush, the females commonly begin incubation after the second egg is laid. Asynchronous hatching, where eggs hatched at different times, is a common occurrence in nests that contain more than two eggs. This hatching asynchrony is particularly prevalent in the high-altitude habitat of these birds.

In this study, we used video monitoring at the nest to record provisioning behaviors of parents and begging behaviors of nestlings. To investigate the reasons behind the size hierarchy among nestlings, we first compared the growth patterns of nestlings of different sizes. Subsequently, we examined the patterns of food allocation to determine whether they were influenced by the parents or the nestlings themselves. Then, we examined the provisioning patterns of parents and factors that influenced the begging intensity of nestlings. Finally, we explored the responses of parents made to the nestling begging behaviors.

## 2. Materials and Methods

### 2.1. Study Area and Population

From 2010 to 2015, our study was conducted in Luqu County, located at 102.5° E and 34.6° N, with an elevation of 3400 m, in Gansu Province, China. This region is characterized by abundant annual precipitation, ranging from 633 to 782 mm, a low annual mean temperature of 2.3 °C, and significant elevation gradients, ranging from 2900 to 4300 m. The natural landscapes changed from spruce forest at a low altitude to alpine meadow at a high altitude, with shrubs dispersing on the grassland that are composed mainly by hemsley’s barberry (*Berberis hemsleyana*), sea buckthorn (*Hippophae rhamnoides*), goat willow (*Salix caprea*), and maximovicz’s peashrub (*Caragana maximovicziana*) [41]. Plain Laughingthrushes prefer to construct nest in the shrubs of *B. hemsleyana* and *H. rhamnoides*. The breeding density reached 0.04 nests/ha in a study area of 1000 ha.

The breeding season of the Plain Laughingthrush spans approximately four months, starting from early May and extending until the end of August. Our nest search activities began in late April, during which we systematically checked the shrubs for nesting sites. The earliest recorded egg-laying date was observed on May 6th, while the latest was noted on July 25th. In most cases, the female Plain Laughingthrush lays one egg per day until the clutch size is complete. However. in some instances, the female may lay the last egg after a one-day interval. The typical clutch size for this species is three eggs, although nests containing two or four eggs have been occasionally recorded. Following the laying of the second egg, females usually commence all-day incubation. However, after laying the first egg, females begin to prevent the eggs from freezing, as the night temperature on the Tibetan Plateau can drop below zero. As a result, hatching asynchrony is a common occurrence within nests that contain more than two eggs. The incubation period for the Plain Laughingthrush eggs is approximately 17 to 19 days, with an average duration of 17.8 days (±0.6, *n* = 34 nests). Once hatched, the nestlings require approximately 11 to 14 days to fledge, with an average duration of 11.7 days (±1.5, *n* = 48 broods). Despite the hatch asynchrony, the nestlings tend to depart from the nests synchronously.

### 2.2. Life History Data Collection

We checked the nest content daily during the time of egg-laying to determine the laying sequence of each egg. We marked the laying sequence on the eggshell using nontoxic marking pens. (Deli Company, Guangzhou, China). Then, the fresh mass (to the nearest 0.1 g, JiuYi Electronic Technology Co., Ltd., BL01, Ningbo, Zhejiang Province, China), length (L), and width (B, to the nearest 0.1 mm, Masterproof International Trade Co., Ltd., 431, Shanghai, China) of each egg were measured. The egg volume (V) was calculated according to the equation: V = Kv × L × B^2^. Here, Kv is the coefficient of 0.507 [42]. Thereafter, nests were inspected daily to monitor incubation and the hatching events. Early-hatched nestlings were darker in color than later-hatched ones, based on which we could exactly determine the hatching sequence of each nestling. We used nontoxic marking pens (the same pens as above) to mark the nestlings on the head for individual identification. For each new hatchlings, the body weight and tarsus length were measured (the same tools as above). There is another point to be made: beak gape begins with tips of mandibles and ends with commissural point. When measuring, place one end of the vernier caliper on tips of mandibles and the other end on the commissural point to read. Thereafter, we measured the body traits of nestlings every two days to monitor their growth until they were preyed upon by predators or fledged.

### 2.3. Behavioral Date Collection

To capture and document the behaviors of adult Plain Laughingthrush and nestlings, we used digital camcorders (ZX1, Eastman Kodak Company, Rochester, NY, USA). Our recording methodology involved positioning a tripod approximately 1 m diagonally above the nest cup during the nest construction phase. To minimize disturbance and ensure accurate observations, we waited for the adult birds to cease their alarm calls directed towards the tripod. Once the adults left the nest area to forage, we mounted the camcorder onto the tripod. Video recording sessions for each nest occurred every two days, specifically between 9:00 AM and 11:00 AM, following China standard time. Throughout the recording process, we refrained from any nest-visiting activities to minimize disturbance and interference with parental behaviors. It is worth noting that no instances of nest desertion occurred as a result of our video recording activities, indicating that the process had no adverse effects on the focused-on individuals. After completing the video recordings, we transferred the footage to a computer for subsequent analysis. By playing back the videos on the computer, we were able to extract behavioral data related to the adult Plain Laughingthrush and nestlings.

The process of re-feeding in the Plain Laughingthrush nest can be viewed as a sequential behavior where prey items are transferred from the one nestling’s mouth to another. The provisioner, typically one of the adult birds, initiates the feeding process by delivering food to a particular nestling. If the receiving nestling does not swallow the food, the provisioner retrieves the food from its mouth and proceeds to deliver it to another nestling within the same nest.

During a feeding bout, which refers to the period from when adult bird arrives at the nest until it departs, several data points can be collected regarding the behavior of adult birds. These data points include the following: (i) Number of provisioners, which refers to the count of adult birds involved in providing food during the feeding bout. (ii) The feeding position of the provisioner (adult birds) delivering the food. The nest is divided into four quadrants, along with a central location, creating five distinct areas. The specific feeding position is determined based on the location of the parental beak during the act of food delivery [43]. (iii) Insect size: the size of the insects provided as food is categorized based on the relative length compared to the beak gape. An insect smaller than the beak gape is denoted as a small insect, an insect equal to the beak gape is classified as a medium insect, and an insect larger than the beak gape is denoted as a large insect. In terms of scoring, a small insect is scored 1, a medium insect is scored 2, and a large insect is scored 3. The number of prey insects held in the beak of the provisioner is recorded. (iv) Re-feeding behavior: it is noted whether the provisioner exhibits re-feeding behavior during the food delivery.

Data about nestling behaviors during feeding bouts include the following: (i) Identity of the nestling that first received food from the provisioner. (ii) Identity of nestling that actually swallowed the food. (iii) Head-raising behavior of the fed nestling: it is noted whether the nestlings raised their heads to beg for food when the adults arrived at the nest. This behavior is recorded as either “yes” (1) or “no” (0). (iv) Head raising order of the fed nestling: the order in which the fed nestlings raised their heads to beg for food is scored. A score of I, ranging from 4 to 1, is assigned to each nestling based on their position in the head-raising sequence, with “4” indicating the first to raise its head and “1” indicating the last. (v) Height that the fed nestling stretched its neck: the height to which the fed nestling extended its neck while begging for food is scored. A score of II, ranging from 4 to 1, is assigned based on the relative height, with “4” representing the highest stretching of the neck and “1” representing the lowest. (vi) Distance between the beak of the provisioner and the fed nestling: the distance between the beak of the adult bird and the fed nestling is measured. A score of III, ranging from 4 to 1, is assigned based on the proximity, with “4” indicating the closest distance and “1” indicating the farthest. (vii) Aggressive behaviors among nestlings: it is recorded whether the nestlings exhibited aggressive behaviors such as pecking or jostling each other during the feeding bout. (viii) Swallowing of food by the first food-receiver: it is noted whether the nestling that was first fed by the provisioner actually swallowed the food provided. (ix) Number of nestlings that obtained food: the count of nestlings that successfully obtained food during the feeding bout is recorded.

Based on the collected data, several estimated were derived. (i) Total provisioning rates: they were estimated by calculating the number of feeding bouts performed by both male and female birds per hour. (ii) Parental feeding efficiency: it was estimated by calculating the proportion of fed nestlings in each feeding bout relative to the total brood size. (iii) Begging intensity of the first food-receiver: it was estimated by averaging the total scores of I–III (head-raising order, neck stretching height, and distance from the provisioner’s beak). These were normalized based on the brood’s size. For two-chick nests, the average score was divided by two; for three-chick nests, it was divided by three; and for four-chick nests, it was divided by four [43].

### 2.4. Statistical Analysis

We conducted generalized linear mixed models to investigate factors influencing egg size, body weight, linear measurements of nestlings, parental provisioning rates, food amount delivered in each feeding bout, parental feeding efficiency, and begging intensity of the first food-receiver. The independent variables included clutch size, laying sequence of each egg (Table 1), brood size, hatching sequence of each nestling, nestling age (Table 2 and Table 3), and total amount of food delivered by the provisioner (Table 4). The dependent variables in all models were normally distributed, so we selected the identity link function. Random effects were included for year and nest identity to account for repeated measurements. In the model selection process, we initially introduced all main effects and interactive effects of the independent variables to create a maximal model. Subsequently, we iteratively removed the least significant effect from the model. Akaike’s information criterion was employed to evaluate model fit, with lower criterion values indicating better fit to the data. We used one-way ANOVA (multiple comparisons by LSD) to examine whether there was size hierarchy within the brood (by comparing the hatching body weights of nestlings with different hatching sequences), and whether nestlings exhibited allometric growth between body weight and linear measurements (by comparing tarsus length and beak gape of nestlings with different hatching sequences, after controlling the body weight through fitting a univariate linear model). The variation of parental provisioning rates in different-sized nests was fitted with nestling age by a univariate linear model. After controlling the nestling age, parental provisioning rates in three-chick nests were compared with that in two- or four-chick nests using one samples *t*-tests. The percentage of feeding bouts in which nestlings with the highest begging intensity became the first food-receiver and in which the first food-receiver swallowed the food was compared with the random value using one-sample *t*-tests, after being transformed using an arcsine square root function.

All analyses were performed using SPSS (version 19.0, IBM, Armonk, NY, USA). Descriptive results are presented as mean ± 1 SE. The null hypotheses were rejected when *p*_two-tailed_ < 0.05. We performed one-sample Kolmogorov–Sminov tests to assess the normality of the data. However, if the data did not meet the criteria for normal distribution, we utilized non-parametric tests instead.

### 2.5. Ethics Statement

The bird measurement procedures were performed in according with the Wildlife Conservation Law established by the tenth National People’s Congress of China on 28 August 2004. Furthermore, our experimental procedures received explicit approval from the Animal Research Ethics Committee of Lanzhou University. To ensure individual identification, the eggs were marked with nontoxic pens on the eggshell, and measurements were taken using a vernier caliper. It is worth noting that these procedures did not appear to have any detrimental effects on the hatching process results.

Females tended to lay larger eggs with the laying sequence (*p* < 0.001; Table 1). The clutch size had no influence on the egg size (*p* = 0.374, Table 1). The body weight of hatchlings differed significantly with their hatching sequences (first: 4.67 ± 0.05 g, *n* = 45; second: 4.75 ± 0.03 g, *n* = 42; third: 4.44 ± 0.10 g, *n* = 44; last: 4.24 ± 0.16, *n* = 7; *F*_3,134_ = 5.14, *p* = 0.002). This difference indicated that size hierarchy within the brood was established within the brood after all nestlings hatched (Figure 1), with the first and second nestlings being equal (*p* = 0.374), and both of them being larger than the third (both *p* ≤ 0.021) and last nestlings (both *p* ≤ 0.024).

Body weight of nestlings significantly increased with nestling age but decreased with their hatching sequences (both *p* ≤ 0.016) and was unrelated to brood size (*p* = 0.982; Table 2). Consequently, the growth pattern of body weight differed significantly among nestlings with different hatching sequences, with those early-hatched nestlings having larger fledgling body weight than later-hatched nestlings (Figure 2A).

Both the linear measurements of nestlings, i.e., tarsus and beak gape, increased significantly with nestling age (both *p* < 0.001) but were unrelated to their hatching sequences (both *p* ≥ 0.538) and brood size (both *p* > 0.111, Table 3). Furthermore, the beak gape decreased significantly with the interplay of brood size and nestling age (*p* = 0.030, Table 3), indicating that the beak gape of nestlings was smaller when there are more nestlings within the nest (Figure 2B).

Both the tarsus and beak gape of nestlings grew linearly with their body weight (all R^2^ > 0.69, *p* < 0.001; Figure 3). After controlling the body weight, nestlings with different hatching sequences had no differences in their tarsus length (*F*_3,58_ = 0.13, *p* = 0.940; Figure 3A); however, they differed significantly in their beak gape (*F*_3,58_ = 5.54, *p* = 0.002; Figure 3B), with the last nestlings having the smallest beak gape within the brood.

By identifying the food items delivered by parents to the brood, seven major food types were listed on the dietary of nestlings, including Lepidoptera adult and larva, Coleoptera adult and larva, Orthoptera adult, Diptera adult and spiders. In 580 feeding bouts in which the food types can be exacted identified, Lepidoptera larva occupied 78.62%, significantly higher than other food items (χ_6_^2^ = 1985.91, *p* < 0.001; Figure 4A). Limited by the top structure of the shrubs in which the nests were constructed, parents generally provisioned the brood at a fixed position (the first feeding position, 82.63 ± 4.01%, *n* = 24 nests), which was significantly higher than any other feeding positions (all *p* < 0.001, Figure 4B).

Male and female both participated in food delivery. The total provisioning rates increased significantly with brood size and nestling age (both *p* ≤ 0.002, Table 4). After controlling the nestling age, the mean provisioning rates of parents in three-chick nests (9.12 ± 0.95, *n* = 11) were equal to that in two-chick nests (10.34 ± 2.10, *n* = 11; *t*_10_ = 2.35, *p* = 0.448) but significantly lower than that in four-chick nests (11.63 ± 1.64, *n* = 11; *t*_10_ = 2.35, *p* = 0.040; Figure 5). The food amount delivered in each feeding bout by parents did not change with nestling age (*p* = 0.223) but increased with brood size (*p* = 0.004; Table 4). Parental feeding efficiency also did not change with nestling age (*p* = 0.124), but increased with brood size and the food amount delivered in each feeding bout (both *p* < 0.001; Table 4).

During the feeding bout, the provisioners exhibited re-feeding behaviors if the first food-receiver did not swallow the food item immediately. In 112 of 307 (36.5%) feeding bouts, the food was taken out of the mouth of the first food-receiver and allocated to another nestling with lower begging intensity. As a result, although the nestling with the highest begging intensity was more likely to become the first food-receiver (two-chick nests: 59.59 ± 2.14%, *n* = 6; *t*_5_ = 4.47, *p* = 0.007; three-chicks nests: 44.91 ± 1.31%, *n* = 11; *t*_10_ = 8.85, *p* < 0.001; four-chicks nests: 38.92 ± 1.74%, *n* = 7; *t*_6_ = 8.01, *p* < 0.001), the food amount that the first food-receiver swallowed was finally less than or equal to the average (two-chick nests: 41.87 ± 1.63%; *t*_5_ = 4.98, *p* = 0.004; three-chicks nests: 27.39 ± 1.33%, *t*_10_ = 4.44, *p* = 0.001; four-chicks nests: 20.68 ± 1.48%, *t*_6_ = 2.93, *p* = 0.026; Figure 6).

No aggressive behaviors, such as pecking and jostling, were observed among nestlings when a provisioner delivered food to the nest. Postural activities were the most important method for nestlings to attract the attention of provisioners. The begging intensity of the first food-receiver increased with nestling age (*p* < 0.043) decreased with brood size (*p* < 0.001), but was unrelated to its hatching sequence and the food amount delivered by the provisioner (both *p* > 0.05; Table 5). These results indicated that (1) nestlings had to increase their begging intensity with the growth to attract more attentions of provisioners; (2) when there were more nestlings within the nest, counter-intuitively, the begging intensity of the first food-receiver decreased; (3) larger nestlings did not always exhibit higher begging intensity, although they were more competitive than their smaller, later-hatched brood-mates.

## 3. Discussion

In the Plain Laughingthrush, a size hierarchy is established when there are more than two nestlings hatched in the nest. This size hierarchy not only influences the growth pattern of nestlings of different sizes, but also affects their begging behaviors and the feeding strategies employed by the parents.

Parental birds typically employ two strategies to create variation in offspring size within a brood. The first strategy is known as the egg-size strategy, where females invest differently to eggs based on their laying sequences [45]. The other strategy is called the asynchronous hatching strategy, where females commence incubation before the entire clutch is complete, resulting in asynchronous hatching of nestlings [46]. In the case of the Plain Laughingthrush, egg sizes increase with the laying sequence, suggesting that later-hatched offspring would be expected to be larger than their elder siblings if they hatched synchronously (Table 1). However, contrary to this expectation, the early-hatched nestlings are actually larger than their younger siblings. Therefore, the establishment of size hierarchy within the brood of the Plain Laughingthrush is attributed to asynchronous hatching rather than variation in egg size (Figure 1).

In the asynchronous nests of altricial birds, younger nestlings face disadvantages when competing for food against their elder siblings [2,3,47,48,49]. When food resources are limited, younger nestlings are often subjected to suppression or even aggression from their elder brood-mates if they attempt to access the preferred food-receiving position [50]. However, in the case of the Plain Laughingthrush, despite the parents frequently delivering food to a predictable position (Figure 4B), larger nestlings were observed, as recorded by digital camcorders, to exhibit no obvious intention to occupy that position and displayed no aggressive behaviors towards smaller nestlings. As a result, the relative positions of the nestlings within the nest did not change after the feeding bout. This indicates that sibling rivalry for food in the Plain Laughingthrush is less intense. Instead, we found that nestlings mainly used postural activities such as gaping and stretching their necks to compete for food.

In small-sized passerine birds, postural activities are the major form of nestling begging for food [43,51]. In broods with intense sibling rivalry, where competition for food is high, nestlings that receive more food or are fed first by provisioners usually show higher intensity of postural activities [29,30,38,51]. In such cases, nestlings with larger linear measurements have an advantage in reaching the best food-receiving position (the sibling rivalry hypothesis [52]) or approaching the provisioners more closely compared to nestlings with smaller linear measurements (the parental approaching hypothesis [17]). To compensate for their lower hierarchy in size, younger nestlings often exhibit allometric growth, where their linear measurements grow faster than their body weight [43,53,54]. However, in the case of Plain Laughingthrush, we found that the hatching sequence only influenced the growth of nestling body weight (Table 2) and not their linear measurements (Table 3). Moreover, even after controlling for body weight, the tarsus length of nestlings was unrelated to their hatching sequences, and last-hatched nestlings had smaller beak gape than their elder siblings (Figure 3). These results indicate that allometric growth between body weight and linear measurements did not occur in the Plain Laughingthrush, providing further evidence that sibling rivalry for food is weak in this species.

Compared to the extensive literatures on food allocation among nestlings by parent birds [13,16,17,19,26,28,43], our study represents the first report of parents employing a re-feeding tactic when satiated nestlings fail to swallow the food. In such cases, the begging behavior of satiated nestlings is interpreted by the parents as dishonest signal of need. This control mechanism exercised by the parents in food allocation can explain why the first food-receivers exhibit counter-intuitive responses to brood size and hatching sequence (Table 5). Since the percentage of food-swallowing among the first food-receivers is, in fact, lower than average due to the re-feeding tactic employed by parents (Figure 6), larger nestlings do not need to compete for more food when they are not hungry.

With the growth of nestlings of the Plain Laughingthrush, their parents increase the provisioning rates. Additionally, with an enlarged brood size, the parents also increase the amount of food delivered in each feeding bout and improve feeding efficiency (Table 4). This feeding strategy, combined with the re-feeding tactic, can enhance the survival of smaller nestlings. When more food is delivered to the nest in each feeding bout and feeding efficiency is improved, the likelihood of smaller nestlings obtaining food are increased. This feeding strategy may be influenced by the food availability in our study area. As shown in Figure 4A, the Plain Laughingthrush in our study area benefits from abundant food resources compared to other geographic populations. This is evident not only from the higher provisioning rates observed in our study area (9–12 bouts/h) compared to other populations (5–8 bouts/h [40]) but also from the greater diversity of food items available (including Lepidoptera adult and larva, and Diptera adult) compared to the more limited food types in other populations [40]. In such a resource-rich environment, raising additional offspring, even if they have lower body condition, can enhance the reproductive success of the parents [55].

Certain phenotypic traits in nestlings, such as plumage badges or UV reflectance measurements, can serve as honest cues of body quality that parents use to adjust their food allocation patterns according to environmental conditions [26,56,57]. In contrast, nestling begging behaviors are often not honest signals of actual need because they are exaggerated when there is high intra-brood competition [48,58]. If provisioners are attracted solely by these exaggerated signals, nestlings that genuinely require food may not receive an adequate share. Unlike other passerines where parents complete the feeding bout by delivering food exclusively to a specific nestling [13,59], the parents of the Plain Laughingthrush employ a re-feeding tactic during feeding. Since nestlings with the highest begging intensity, typically the early-hatched and larger nestlings, are more likely to be the first to receive food, it indicates that parents cannot discern which nestlings genuinely need the food based solely on their postural activities. However, if the initial food-receivers are not particularly hungry and do not immediately swallow the food, the parents retrieve the food and reallocate it to a nestling with lower begging intensity than the first recipient. Thus, the response of the first food-receiver to the food serves as an honest signal of the nestling’s actual need for food. Parents can use this signal to determine their re-feeding tactic, thereby reducing the likelihood of larger nestlings monopolizing the food.

When Plain Laughingthrushes breed on the Tibetan Plateau and produce more than two offspring in the nest, parent birds tend to establish a size hierarchy among nestlings. Nestlings with higher hierarchy possess greater competitive abilities in attracting the attention of provisioners, and thus their begging behaviors may influence the primary food allocation by the parents. However, when satiated nestlings do not immediately swallow the food they receive, parents re-allocate the food to another nestling with lower competitive ability. This behavior indicates that parents are able to discern dishonest signals of need based on whether the food-receivers actually consume the food. By employing this re-feeding tactic, parents effectively suppress the attempts of larger nestlings to monopolize the food and ensure the survival of smaller nestlings.

## 4. Conclusions

Due to hatching asynchrony, larger nestlings have a competitive advantage in food acquisition over their smaller brood-mates; however, they did not always exhibit higher begging intensity than the latter. Generally, nestlings with the highest begging intensity were more likely to be fed first, indicating that nestling begging played a key role in determining parental food allocation. However, if the first food-receivers were satiated and did not swallow the food immediately, parents picked up the food and re-allocated it to another nestling. This re-feeding tactic of parents reduced the possibility that early-hatched nestlings monopolized the food by taking advantage of their larger body size. To date, our research demonstrates that while parental food allocation primarily hinges on the begging intensity of the nestlings, the decision to re-feed is contingent upon whether the initial recipients of the food ingest it immediately.

## Figures and Tables

**Figure 1 animals-13-03522-f001:**
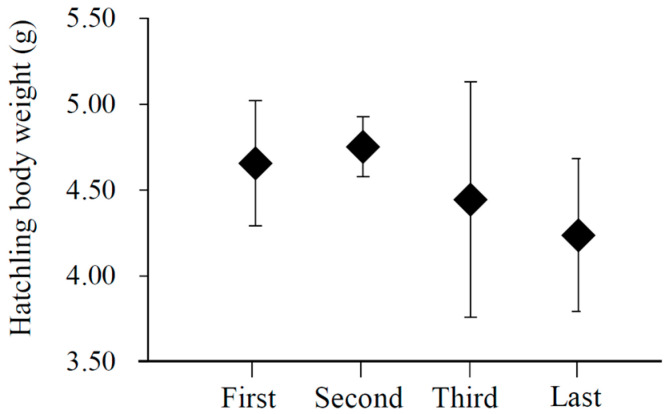
The relationship between hatchling body weight and the hatching sequence of nestlings in the Plain Laughingthrush that breed on the Tibetan Plateau, the points represent the means, and the vertical lines represent the standard deviations.

**Figure 2 animals-13-03522-f002:**
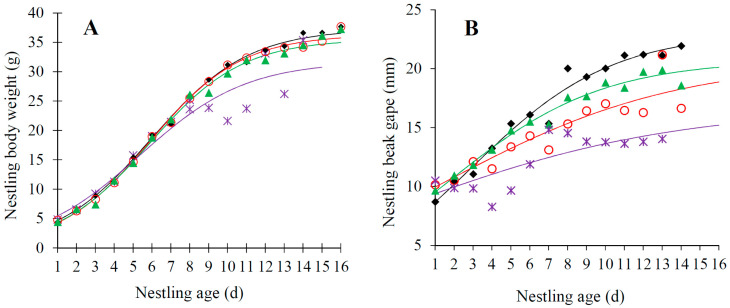
Variation in nestling body weight (**A**): first nestlings, diamonds and black line; second nestlings, circles and red line; third nestlings, triangles and green line; last nestlings, stars and grape line. Beak gape (**B**): one-chick nests, diamonds and black line; two-chick nests, circles and red line; three-chick nests, triangles and green line; four-chick nests, stars and grape line. Nestling age in the Plain Laughingthrush that breed on the Tibetan Plateau.

**Figure 3 animals-13-03522-f003:**
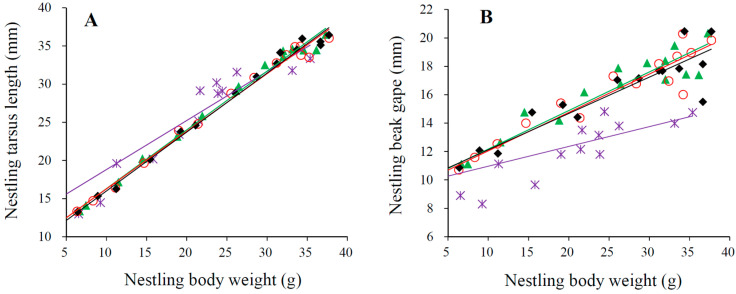
The growth pattern of tarsus (**A**) and beak gape (**B**) with the body weight of nestlings with different hatching sequences (first nestlings, diamonds and black line; second nestlings, circles and red line; third nestlings, triangles and green line; last nestlings, stars and grape line), examined by linear regression.

**Figure 4 animals-13-03522-f004:**
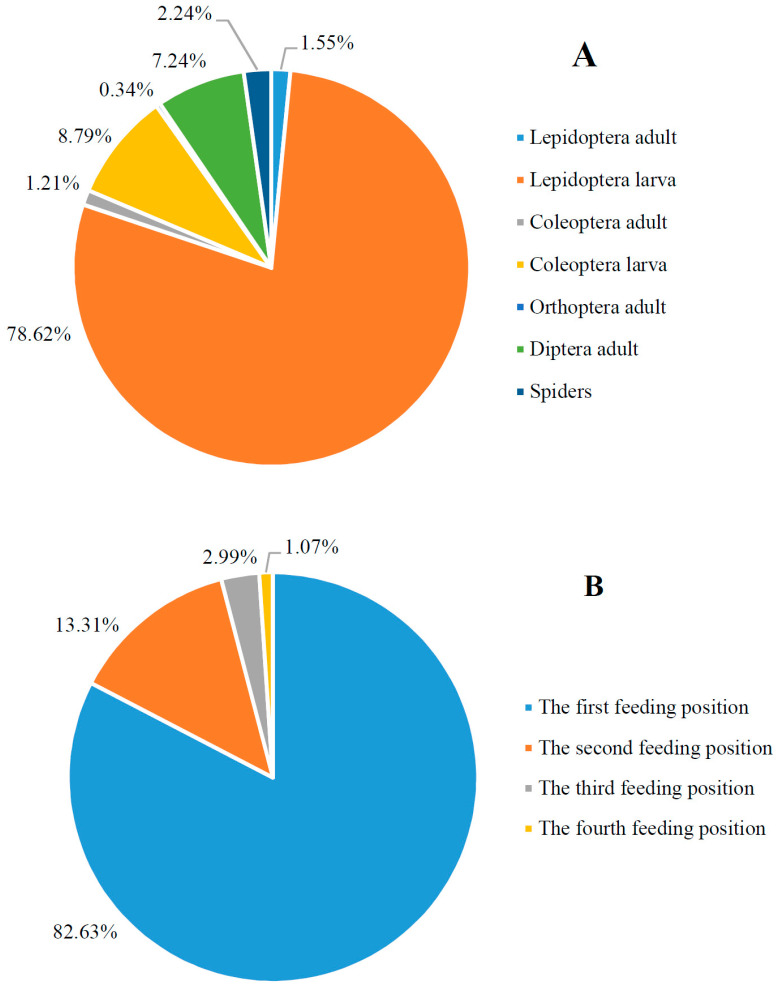
The percentage of different nestling dietary items delivered to the brood by the parents of Plain Laughingthrush (**A**) and that of their feeding positions (**B**) on the Tibetan Plateau.

**Figure 5 animals-13-03522-f005:**
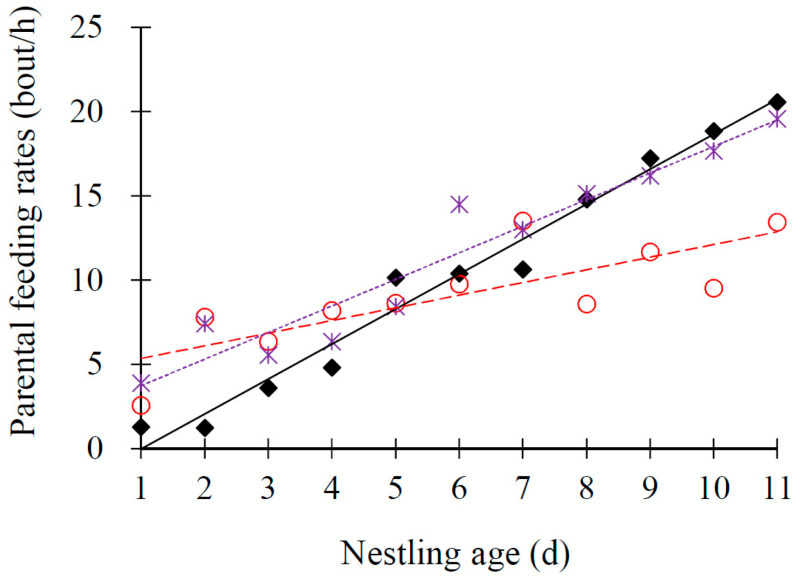
Variation in parental feeding rates with nestling age in the Plain Laughingthrush that breed on the Tibetan Plateau (two-chick nests, diamonds and black line; three-chick nests, circles and red line; four-chick nests, stars and grape line).

**Figure 6 animals-13-03522-f006:**
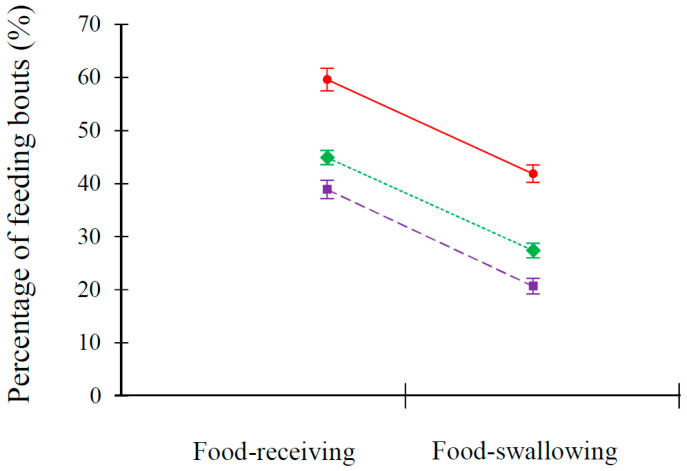
The percentage of feeding bouts in which the nestlings with the highest begging intensity become the first food-receiver (food-receiving) and the first food-receivers swallow the food (food-swallowing), red represents two-chick nests, green represents three-chick nests, and purple represents four-chick nests.

**Table 1 animals-13-03522-t001:** The influence of clutch size and laying sequence on the egg size (indexed by fresh mass) of the Plain Laughingthrush, examined by fitting a generalized linear mixed model (year and nest identity were set as random effect variables).

Statistical Results Fixed Effect Variables	*β* ± SE	t	*n*	*p*	90% CI
Intercept	5.15 ± 0.31	16.55	72	<0.001	4.54–5.76
Clutch size	0.09 ± 0.10	0.89	72	0.374	−0.11–0.29
Laying sequence	0.06 ± 0.02	3.72	72	<0.001	0.03–0.10

**Table 2 animals-13-03522-t002:** Factors that may influence the body weight of nestlings of the Plain Laughingthrush, examined by fitting generalized linear mixed models (year and nest identity were set as random effect variables).

Statistical Results Fixed Effect Variables	*β* ± SE	t	*n*	*p*	90% CI
Brood size	−0.01 ± 0.24	−0.02	50	0.982	−0.48–0.47
Hatching sequence	−0.41 ± 0.17	−2.42	50	0.016	−0.74–−0.08
Nestling age	2.61 ± 0.04	74.32	50	<0.001	2.55–2.68

The growth curves in body weight and beak gape of nestlings were fitted by logistic models: *W* = *K*/ (1 + exp (*a* − *b* × *t*)), where ‘*K*’ is the asymptotic body weight that nestlings reach at fledging and ‘*a*’ is the time when nestlings begin to grow.

**Table 3 animals-13-03522-t003:** Factors that may influence the linear measurements of nestlings of the Plain Laughingthrush, examined by fitting generalized linear mixed models (year and nest identity were set as random effect variables). The asterisk (*) symbol represents the interaction between random factors in the model.

Statistical Results Fixed Effect Variables	*β* ± SE	t	*n*	*p*	90% CI
Nestling tarsus length					
Intercept	10.02 ± 0.86	11.66	50	<0.001	8.33–11.70
Brood size	0.03 ± 0.29	0.11	50	0.916	−0.54–0.60
Hatching sequence	−0.08 ± 0.13	−0.62	50	0.538	−0.34–0.18
Nestling age	2.09 ± 0.03	69.04	50	<0.001	2.03–2.15
Nestling beak length					
Intercept	8.24 ± 1.22	6.78	50	<0.001	5.86–10.63
Brood size	0.37 ± 0.23	1.60	50	0.111	−0.09–0.83
Hatching sequence	−0.01 ± 0.07	−0.16	50	0.872	−0.14–0.12
Nestling age	0.99 ± 0.08	12.37	50	<0.001	0.83–1.15
Brood size * nestling age	−0.06 ± 0.03	−2.18	50	0.030	−0.11–−0.01

**Table 4 animals-13-03522-t004:** Factors that may influence total provisioning rates, food amount delivered in each feeding bout and parental feeding efficiency of the Plain Laughingthrush, examined by fitting generalized linear mixed models (year and nest identity were set as random effect variables).

Statistical Results Fixed Effect Variables	*β* ± SE	t	*n*	*p*	90% CI
Parental total provisioning rate					
Brood size	0.67 ± 0.21	3.20	24	0.002	0.26–1.09
Nestling age	1.41 ± 0.09	15.25	24	<0.001	1.23–1.60
Food amount delivered in each feeding bout					
Brood size	1.70 ± 0.59	2.89	24	0.004	0.55–2.86
Nestling age	0.06 ± 0.05	1.22	24	0.223	−0.03–0.15
Parental feeding efficiency					
Brood size	0.11 ± 0.01	11.55	24	<0.001	0.09–0.13
Nestling age	−0.01 ± 0.01	−1.54	24	0.124	−0.01–0.01
Food amount delivered in each feeding bout	0.02 ± 0.01	10.69	24	<0.001	0.02–0.03

**Table 5 animals-13-03522-t005:** Factors that may influence the begging intensity of the first-food receiver in each feeding bout, examined by fitting GLMMs (year and nest identity was set as random effect variables).

Statistical Results Fixed Effect Variables	*β* ± SE	t	*n*	*p*	90% CI
Intercept	2.11 ± 0.15	13.86	24	<0.001	1.81–2.41
Brood size	−0.22 ± 0.06	−3.89	24	<0.001	−0.33–−0.11
Hatching sequence	−0.02 ± 0.03	−0.72	24	0.470	−0.08–0.04
Nestling age	0.02 ± 0.01	2.03	24	0.043	0.01–0.04
Food amount delivered	−0.01 ± 0.01	−0.74	24	0.461	−0.01–0.01

Exponentially, ‘*b*’ is a constant scaling for growth rate, and ‘*t*’ is nestling age [44].

## Data Availability

The data presented in this study are available on request from the corresponding author. The data are not publicly available due to privacy.

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
