# Peer review of "Breeding Behavior, Nestling Growth, and Begging Behavior in the Plain Laughingthrush (*Garrulax davidi*): Implications for Parent–Offspring Conflict"

_animals, 2023, doi:10.3390/ani13223522_

Round 1

Reviewer 1 Report

Comments and Suggestions for Authors

A few suggestions (see attached file). My biggest concern is the extra information in the results that have not been mentioned in the introduction or methods.

Overall, I found this a very interesting study.

Comments on the Quality of English Language

As I imagine none of the authors are native English speakers, the quality of the English is very good.  Just some moderate editorial tweaks are required.

Author Response

To Reviewer 1

Thank you for your insightful feedback on the manuscript. Based on your recommendations, we have made the necessary revisions to the relevant sections of the paper. At the same time, experts in the field were invited to polish the article to improve the level of English expression. Some of your questions were answered below.

General comments:

Whilst I found this an interesting article, I feel that it could be better served by being developed into two manuscripts. There is information introduced that seemed extraneous to the central theme.  I also think that there needs to be a reconsideration of the methods and the results sections.

However, that being said, this article attempts to fill an obvious knowledge gap and certainly left me wanting to know more.

Specific comments

Abstract:

The abstract is concise and covers the salient points of the results well.

Introduction: 

A good background to the study is given and it is placed in context with related research. There are some readability issues, but this is probably due to the authors not being native-English speakers, and it is a minor concern.

Line 91 – 93: Rearrange these sentences; maybe start with “To explore the reasons that caused size hierarchy among nestlings we…”

Based on your comments, we have revised the sentence “To investigate the reasons behind the size hierarchy among nestlings, we first compared the growth patterns of nestlings of different sizes. Subsequently, we examined the patterns of food allocation to determine whether they were influenced by the parents or the nestlings themselves.”

Methods & Materials:

This section is clear, however, tables 1-5 would be better situated within the results section.  The choice of statistical analyses is appropriate.

According to your advice, we have adjusted the position of the table.

Results:

This is where I got a bit confused. The results, in the main, correspond with the questions posed in the introduction; however, the authors include data regarding food types that were not mentioned in the introduction or methods section except concerning insect size. I think this should either be removed (and, with analysis, could certainly provide another manuscript (see Höhn et al 2023 as an example)), or a brief discussion of food types, availability and relevance should be added to the introduction, and again mentioned in the methods section.

I particularly liked the figures presented in the results section; they were clear and easy to interpret.

Thank you for the reviewer's suggestions. We have provided a detailed description of the mentioned content in Figure 4, showcasing the types of food and their proportions.

Discussion:

I found the discussion a pleasure to read.  It flows well and sums up the research accordingly.  The study is appropriately discussed and compared with similar research. 

Reviewer 2 Report

Comments and Suggestions for Authors

This study looked at food allocation and aspects of POC in the plain laughingthrush. They analyzed feeding behaviors from the parents and begging behaviors from the nestlings to ascertain how hatching sequence and body size may influence food allocation.

Line 40 -42: Rewrite this sentence, something is missing or not needed.

Line 51: should it be largest-size young?

Line 52: rephrase, “To date demonstration the relative…”

Line 80: We sought to address/answer these questions in the plain laughingthrush.”

Common names do not need to be capitalized unless they are a specific place or names after someone.  

Line 105: Are the specific shrub species needed can this be more general or even coupled with the common names?

Line 107:

Line 111: change at to on

Lines 123-124: These sentences are in the passive voice; can you update then to the active. We collected… we marked…

Line 136-137: It is unclear what the point is they are making.

Line 148: we did not observe any nest desertion…

Line 176-181: This section reads more like something for the analysis section.

Line 236: What nonparametric tests did you use?

Line 245: It may not be needed to restate the bird type so often since there is only one study species.

Figure 4: I would consider a bar graph for the visualization.

The discussion is very through and clearly describe their data along with the literature.

Comments on the Quality of English Language

The writing is very clear. There are one or two places where the language could be clearer and the authors should avoid using the passive voice when possible. 

Author Response

To Reviewer 2:

I am very grateful to your comments for the manuscript. According with your advice, we amended the relevant part in the manuscript. At the same time, experts in the field were invited to polish the article to improve the level of English expression. Some of your questions were answered below. 

Line 40 -42: Rewrite this sentence, something is missing or not needed.

Answer: Based on your comments, we have revised the sentence “The question of who determines the food allocation pattern within the brood has long been addressed for altricial birds with multiple young in one breeding attempt.”

Line 51: should it be largest-size young?

Answer: In this context, it's not the largest nestling but rather the larger nestlings, as both larger and smaller nestlings compete with each other, not just the largest ones competing with the smaller ones.

Line 52: rephrase, “To date demonstration the relative…”

Answer: As required, we have rephrase it: “To date, elucidating the relative contributions of parents and offspring in determining food distribution remains a crucial endeavor for understanding the evolution of Parent-Offspring Conflict (POC) in animal families.“”

Line 80: We sought to address/answer these questions in the plain laughingthrush.”

Common names do not need to be capitalized unless they are a specific place or names after someone.  

Answer: Yes, we have made the revisions.

Line 105: Are the specific shrub species needed can this be more general or even coupled with the common names?

Answer: According to your advice, we have made revisions to this section: “hemsley's barberry (Berberis hemsleyana), sea buckthorn (Hippophae rhamnoides), goat willow (Salix caprea) and maximovicz's peashrub (Caragana maximovicziana)”

Line 111: change at to on

Answer: In response to your questions, we have made revisions to that section.

Lines 123-124: These sentences are in the passive voice; can you update then to the active. We collected… we marked…

Answer: We have made revisions to this section: “We checked the nest content daily during the time of egg-laying to determine the laying sequence of each egg. We marked the laying sequence on the eggshell using nontoxic marking pens.”

Line 136-137: It is unclear what the point is they are making.

Answer: In this sentence, we have outlined the detailed procedure for measuring nestling beak gape. “When measuring, place one end of the vernier caliper on tips of mandibles and the other end on commissural point to read.”

Line 148: we did not observe any nest desertion…

Answer: This sentence indicates the impact of video recording on parental care of nestlings. Placing it here is appropriate.

Line 176-181: This section reads more like something for the analysis section.

Answer: This section describes how to calculate and estimate based on the extracted data. These calculation methods provide a clear context for readers to understand the analysis process. Therefore, it's best to place this section in the "Methods" part of the paper.

Line 236: What nonparametric tests did you use?

Answer: The non-parametric test we used is the Wilcoxon Signed-Rank Test.

Line 245: It may not be needed to restate the bird type so often since there is only one study species.

Answer: Thanks for the comments, we have made revision “Females tended to lay larger eggs with the laying sequence (P < 0.001; Table 1).”

Figure 4: I would consider a bar graph for the visualization.

Answer: This is a valuable suggestion, but we believe that using a pie chart to represent the proportions is also appropriate.

Reviewer 3 Report

Comments and Suggestions for Authors

The study by Jinyuan et al. aims to better understand parent-offspring conflict regarding food allocation within broods. To do this, they studied the Plain Laughingthrush and quantified a range of estimates of chick growth, begging intensity, and parent food allocation. The authors found that nestlings that begged more intensely were more likely to be fed first, but if the nestling did not immediately swallow the food then parents re-allocated it to another nestling. The authors conclude that not swallowing food immediately, but not begging intensity, is an honest signal of a nestlings need for food. Overall, the study is well designed and the manuscript is well written. I have some comments below that I hope improve the manuscript.

My primary comment is that I was surprised to see the nestling begging behaviors were not shown in any figures. I realize that figure 6 presents data related to this, but given that the introduction sets up begging intensity being one of the central behaviors in the parent-offspring conflict it seems appropriate to me to present these data. It would allow the readers to see how much variation is present in these behaviors. Based on Table 5, brood size and nestling age are significantly related to begging intensity. Graphs showing these relationships would be great additions. I am also interested in whether the data on likelihood to receive the food first (lines 319 – 322) would be better presented as a graph? Finally, when adults re-allocated food, did the food always go to the next most intense begger? These data would be interesting too and shed more light on whether adults pay attention to begging intensity.

Furthermore, the P-values reported in the results text (lines 328-331) do not appear to match the P-values reported in Table 5.

My second major comment is that the introduction does not include a clear hypothesis or predictions. These would both help readers to understand and evaluate the work.

Minor comments

Line 132: Can the authors provide more detail on how they marked each chick? Did they mark with different colors or different shapes on each chick? I am wondering if it was possible that the adult birds could see these marks, thereby inadvertently influencing their behavior.

Line 170: How did the authors estimate the height a nestling stretched its neck and did they validate that their measures were accurate and repeatable?

Line 227: The authors state they used paired-samples t-tests. Can they explain why they believe this is the most appropriate t-test type for this data. If I'm understanding this correctly, I don't believe that paired-samples t-test is the most appropriate. There is no factor linking two- three- or four-chick nests that warrants the paired test. Independent samples would seem more appropriate because each nest is independent.

Figures: For all figures, the description should state whether the points are averages (or something else) and what the error bars depict (standard deviation, SEM, etc.). Finally, the figures should not use red and green for different groups to improve accessibility for those with color blindness. Blue instead of green would work.

In figure 6, what are the three different colors? I do not see a key to explain this.

Line 420, the authors claim that parents treat begging behavior as a dishonest signal. I suggest the authors be more cautious with this conclusion. It is one possibility, but it could also be that the signal adults use is whether food is swallowed immediately or not, which would be an honest signal. The authors say this later in the discussion, but it should be stated here too.

Comments on the Quality of English Language

In general, the english is excellent, but there are a few instances where the grammar or word choice is questionable so a round of edits would be an improvement.

Author Response

To Reviewer 3

Thanks very much for taking your time to review this manuscript. Your comments were very detailed, and I really appreciate all your comments and suggestions! We have improved the English language and refined the Methods and Results section. Our answers to your points are as follows.

Reviewer Comment: My primary comment is that I was surprised to see the nestling begging behaviors were not shown in any figures. I realize that figure 6 presents data related to this, but given that the introduction sets up begging intensity being one of the central behaviors in the parent-offspring conflict it seems appropriate to me to present these data. It would allow the readers to see how much variation is present in these behaviors. Based on Table 5, brood size and nestling age are significantly related to begging intensity. Graphs showing these relationships would be great additions. I am also interested in whether the data on likelihood to receive the food first (lines 319 – 322) would be better presented as a graph? Finally, when adults re-allocated food, did the food always go to the next most intense begger? These data would be interesting too and shed more light on whether adults pay attention to begging intensity.

Answer: The reviewer's suggestion is appreciated; however, we believe that the table sufficiently demonstrates the relationship between the two variables. “Based on Table 5, brood size and nestling age are significantly related to begging intensity. Graphs showing these relationships would be great additions.” Regarding the question raised by the reviewer, "receive the food first (lines 319 – 322) would be better presented as a graph," we have presented these data in Figure 6. We appreciate the reviewer's inquiry about whether "when adults re-allocated food, did the food always go to the next most intense beggar." In this article, we did not directly analyze this aspect, but we consider further exploration in subsequent experimental analyses. Typically, when parent birds assess honest signals from nestlings during a single feeding bout, they will initially give the food to the nestling with the highest begging intensity. If a nestling is satiated and does not promptly swallow the food, the parent bird will interpret it as a false signal and engage in secondary food allocation.

Furthermore, the P-values reported in the results text (lines 328-331) do not appear to match the P-values reported in Table 5.

Answer: Based on your comments, we have revised it “Postural activities were the most important method for nestlings to attract the atten-tion of provisioners. The begging intensity of the first food-receiver increased with nes-tling age (P < 0.001043) decreased with brood size (P = 0.043< 0.001), but was unrelated to its hatching sequence and the food amount delivered by the provisioner (both P ≤ > 0.04305; Table 5).”

My second major comment is that the introduction does not include a clear hypothesis or predictions. These would both help readers to understand and evaluate the work.

Answer: Because this is our initial exploration, we were not certain about the factors influencing parental food reallocation. As a result, we did not propose hypotheses or predictions but instead presented the questions we aimed to investigate. Additionally, our research approach is detailed in the last paragraph of the Introduction. I believe that if readers carefully read this section, they should understand the questions we intended to explore: "We first investigate the reasons leading to size differences among nestlings within the nest. Next, we examine whether the food allocation pattern is determined by parents or nestlings. Subsequently, we study parental provisioning patterns and factors influencing nestling begging intensity. Finally, we explore parental responses to nestling begging behaviors."

Minor comments

Line 132: Can the authors provide more detail on how they marked each chick? Did they mark with different colors or different shapes on each chick? I am wondering if it was possible that the adult birds could see these marks, thereby inadvertently influencing their behavior.

Answer: Our marking procedure was as follows: After the nestlings hatched, we used a marker pen to label them on the head and tarsus in hatching order. We did not observe significant behavioral changes in the parents after marking the nestlings. The marking procedure did not appear to have a significant impact on parental care.

Line 170: How did the authors estimate the height a nestling stretched its neck and did they validate that their measures were accurate and repeatable?

Answer: We assessed this through video observations, When the nestling's neck is fully extended, it is scored as 4; if somewhat extended, it is scored as 3. When slightly retracted, it is scored as 2, and when fully retracted, it is scored as 1.

Line 227: The authors state they used paired-samples t-tests. Can they explain why they believe this is the most appropriate t-test type for this data. If I'm understanding this correctly, I don't believe that paired-samples t-test is the most appropriate. There is no factor linking two- three- or four-chick nests that warrants the paired test. Independent samples would seem more appropriate because each nest is independent.

Answer: Yes, the reviewer's suggestion is correct, and we have made the necessary corrections.

Figures: For all figures, the description should state whether the points are averages (or something else) and what the error bars depict (standard deviation, SEM, etc.). Finally, the figures should not use red and green for different groups to improve accessibility for those with color blindness. Blue instead of green would work.

Answer: The points represent the means, and the vertical lines represent the standard deviations. We have provided additional explanations in the figure captions. Thank you for your suggestion regarding the colors in the figures.

In figure 6, what are the three different colors? I do not see a key to explain this.

In lines 341-344, an explanation is provided, indicating the distinctions between two-chick, three-chick, and four-chick nests. Red represents two-chick nests, green represents three-chick nests, and purple represents four-chick nests. We have added this explanation to the figure caption to clarify.

Line 420, the authors claim that parents treat begging behavior as a dishonest signal. I suggest the authors be more cautious with this conclusion. It is one possibility, but it could also be that the signal adults use is whether food is swallowed immediately or not, which would be an honest signal. The authors say this later in the discussion, but it should be stated here too.

Answer: Thanks for the advice, we intend to convey that parental feeding occurs multiple times throughout the day. During the initial feeding each day, nestlings' begging signals are honest when they are hungry. However, if nestlings are satiated and exhibit intense begging behavior without immediately consuming the food during subsequent feedings, this is recognized by the parents as a dishonest signal.